# The Screening and Identification of a Dextranase-Secreting Marine Actinmycete *Saccharomonospora* sp. K1 and Study of Its Enzymatic Characteristics

**DOI:** 10.3390/md22020069

**Published:** 2024-01-28

**Authors:** Boyan Wang, Yizhuo Wu, Qiang Li, Xudong Wu, Xinxin Kang, Lei Zhang, Mingsheng Lyu, Shujun Wang

**Affiliations:** 1Jiangsu Key Laboratory of Marine Bioresources and Environment/Jiangsu Key Laboratory of Marine Biotechnology, Jiangsu Ocean University, Lianyungang 222005, China; boyanwang@jou.edu.cn (B.W.); yizhuowu@jou.edu.cn (Y.W.); qiangli@jou.edu.cn (Q.L.); xudwu@jou.edu.cn (X.W.); leizhang@jou.edu.cn (L.Z.); mslyu@jou.edu.cn (M.L.); 2Co-Innovation Center of Jiangsu Marine Bio-Industry Technology, Jiangsu Ocean University, Lianyungang 222005, China

**Keywords:** dextranase, *Saccharomonospora* sp., characteristics, isooligomaltosaccharides, biofilm, porous starch

## Abstract

In this study, an actinomycete was isolated from sea mud. The strain K1 was identified as *Saccharomonospora* sp. by 16S rDNA. The optimal enzyme production temperature, initial pH, time, and concentration of the inducer of this actinomycete strain K1 were 37 °C, pH 8.5, 72 h, and 2% dextran T20 of medium, respectively. Dextranase from strain K1 exhibited maximum activity at 8.5 pH and 50 °C. The molecular weight of the enzyme was <10 kDa. The metal ions Sr^2+^ and ^K+^ enhanced its activity, whereas Fe^3+^ and Co^2+^ had an opposite effect. In addition, high-performance liquid chromatography showed that dextran was mainly hydrolyzed to isomaltoheptose and isomaltopentaose. Also, it could effectively remove biofilms of *Streptococcus mutans*. Furthermore, it could be used to prepare porous sweet potato starch. This is the first time a dextranase-producing actinomycete strain was screened from marine samples.

## 1. Introduction

Dextranase (E.C.3.2.1.11), which can hydrolyze α-1,6-glucosidic bond, is produced by numerous microorganisms, such as molds, yeasts, and bacteria [1,2,3]. The enzyme is crucial for producing dextran and its derivative products, as well as for sugar, food, and the pharmaceutical industries [4]. Dextranase can hydrolyze high-molecular-weight dextrose to its low-molecular-weight form, giving it antithrombotic properties and serving as a substitute for plasma in the pharmaceutical industry [5]. Moreover, *Streptococcus mutans* biofilms could be removed using dextranase, thereby preventing dental caries [6].

Dextranases are mainly produced by terrestrial microorganisms and are stable under acidic conditions. Microorganisms screened from marine samples can secrete dextranases, and they can exhibit higher activity under alkaline conditions [7,8,9]. To our knowledge, a dextranase produced by a marine actinomycete has not been reported.

Actinomycetes are among the most dominant phylum in the natural environment [10]. Compared with terrestrial bacteria, marine microorganisms are adapted to the marine environment, and their products usually exhibit different characteristics [11,12]. Purushe reported that actinomycete NK458 screened from soil could produce dextranase [13]. Dextranase produced by marine microorganisms exhibited an excellent thermal stability, salt tolerance, and low temperature resistance [14,15,16].

Starch is a polymeric carbohydrate consisting of straight-chain (α-1,4-glucosidic bond) starch and branched-chain (α-1,6-glucosidic bond) starch. Starchy foods differ in their internal biological properties and the composition of their chemical arrangement, which is thought to be a determinant of glucose release during digestion. The products of starch at later stages of processing are mainly classified as rapidly digested starch (RDS), slowly digested starch (SDS), and resistant starch (RS) [17]. Porous starch is a new type of organic adsorbent and embedding material, and it has received widespread attention [18].

In this study, a marine actinomycete *Saccharomonospora* sp. K1 was screened from sea mud. The characteristics of dextranase and its application were investigated. It is a low-molecular-weight enzyme protein and has a high content of oligosaccharides with a high degree of polymerization (DP) in hydrolysates. Moreover, not only can it remove biofilms of *Streptococcus mutans*, but it could also be used to prepare porous sweet potato starch.

## 2. Results

### 2.1. Screening and Identification of Strains

The actinomycete strain isolated from sea mud exhibited dextranase activity on the blue dextran medium (Figure 1A). The mycelia and sporophyte could be observed through SEM (Figure 1B). The strain K1 genome was extracted for PCR amplification, and the amplified fragment was approximately 1500 bp (Figure 1C). The 16S rDNA gene of strain K1 was sequenced. The 16S rDNA sequence was blasted in NCBI, and a phylogenetic tree was drawn using MEGA software ver. 7.0 and Clustal ver. 1.83. The phylogenetic tree indicated that strain K1 belongs to the genus *Saccharomonospora* (Figure 1D).

### 2.2. Fermentation Condition of Dextranase

The effects of seed incubation time and optimal inoculum on the dextranase production by strain K1 are shown in Figure 2A. The optimum seed incubation time and inoculum were 48 h and 10%. The highest enzyme activity was achieved at 72 h (Figure 2B). The optimum temperature for dextranase production by strain K1 was 37 °C, and a higher temperature had a dramatic effect on enzyme production (Figure 2C). The initial pH of the fermentation medium was adjusted from 6 to 11, and dextranase production was most favored at pH 8.5 (Figure 2D) after 72 h of fermentation.

Figure 2E shows that the dextranase activity was the highest when the NaCl concentration was 6 g/L. Strain K1 could tolerate a higher NaCl concentration. When NaCl was 14 g/L, the strain still could produce 70% relative enzyme activity. Moreover, enzyme production was the highest when the liquid content was 25 mL/100 mL flask (Figure 2F).

The effect of different carbon sources on the enzyme production by strain K1 is shown in Figure 3A. Soluble starch was the best carbon source for enzyme production, followed by yeast extract. Similarly, fish powder was the best nitrogen source, followed by potassium nitrate and soybean meal (Figure 3B). Ammonium chloride, ammonium sulfate, sodium nitrate, and tryptone had low effects on enzyme production. Dextranase is an inducible enzyme. The best inducer for strain K1 was 1% dextrose T20 (Figure 3C).

### 2.3. Characteristics of Dextranase

The properties of enzymes affect their application. We prepared a dextranase solution using centrifugation and ultrafiltration and studied their properties.

#### 2.3.1. Effect of Temperature and pH on the Activity of Dextranase

The effect of pH on dextranase activity is shown in Figure 4A. The optimum pH for the maximum enzyme activity was 8.5. The optimal pH for marine dextranases is mostly alkaline conditions, whereas dextranases of terrestrial origin are the opposite [19]. Dextranase from strain K1 maintained more than 60% activity in the pH range of 7.5–9.5 when stored at 25 °C for 1 h, and would exhibit greater activity under alkaline conditions. The aforementioned results revealed that our dextranase had a low pH sensitivity and was suitable for the alkaline condition. The dextranase maintained high activity between 40 and 50 °C (Figure 4B), and the optimum temperature was 50 °C. Regarding the thermal stability of dextranase, this was maintained at almost 100% residual activity for 5 h at 50 °C (pH 8.5), with a 50% enzyme activity loss after 5 h of storage at 55 °C (Figure 4C).

#### 2.3.2. Molecular Weight of Dextranase

The results showed transparent bands below 9.5 kDa (Figure 5), indicating that the molecular weight of dextranase was approximately 9.5 kDa. Most studies have reported that the molecular weight of dextranase is in the range of 60–114 kDa [20,21]. Dextranases produced by fungi also have a molecular weight of at least 23 kDa [22]. According to our knowledge, dextranase produced by strain K1 had the lowest molecular weight so far.

#### 2.3.3. Effect of Metal Ions and Substrate Specificity

Table 1 presents the effects of metal ions on enzyme activity. The enzyme activity was promoted by 10 mM Sr^2+^ and K^+^ treatment, from 100% to 124.25% and 112.62%, respectively, which is similar to the results of Wu et al. [23,24,25,26]. Fe^3+^ and Co^2+^ had a strong inhibitory effect on dextranase activity. Metal ions such as Ca^2+^, Zn^2+^, Ba^2+^, and Mg^2+^ had no significant inhibitory effect.

The substrate specificity of dextranase is shown in (Table 2). Dextranase could specially hydrolyze the α-1,6-glucosidic bond. The most favored substrate was dextrose T20. Meanwhile, the extracted dextranase could hydrolyze the branch chain α-1,6-glucosidic bond in starch. Soluble starch and chitin are mainly composed of α or β-1,4-glucosidic bonds that cannot be hydrolyzed by dextranase.

#### 2.3.4. Analysis of Hydrolysates

The HPLC results showed that isomaltoheptose and isomaltopentaose were the main hydrolysates of dextranase (Figure 6). The main hydrolysate of dextranase reported by Wei Ren et al. was isomalto-oligosaccharide, with a small amount of glucose [27]. The composition of the oligosaccharides was significantly different from that in strain K1 dextranase [26,28]. The peak areas of hydrolysis products were quantified through HPLC by using the Empower GPC software (Waters, Milford, MA, USA) (Table 3). When the hydrolysis reaction time was extended from 0.5 to 2 h, the amount of isomaltoheptaose increased slightly and the percentage was over 97%. Isomaltopentaose was the main product after 6 h of hydrolysis, and its percentage could be reached over 22%. Our results are similar to those that reported the production of a high content of isomalto-oligosaccharides and low quantity of glucose [29]. Isomalto-oligosaccharides could promote the growth and value-adding of human intestinal probiotics and have great commercial value as prebiotics [2,30,31,32]. The high degree of polymerization of isomalto-oligosaccharides has a higher prebiotic effect. Dextranase produced by strain K1 can be used to produce high DP isomaltoheptaose and isomaltopentaose in future.

#### 2.3.5. Effects of Dextranase on Biofilm

The MBIC and MBRC of dextranase were determined (Table 4). Significant effects were observed in the removal of the biofilm combined with the concentration. The plaque inhibition rates were 52.15% and 94.23% when the dextranase concentrations were 4 and 10 U/mL, respectively. The plaque removal rates were 50.12% and 92.54% when the dextranase concentrations were 2 and 10 U/mL. The results of SEM are presented in Figure 7. The biofilm covered the entire slide, and cellular structures closely overlapped when dextranase was absent. As the dextranase concentration increased, the biofilm became progressively thinner and the amount of bacteria decreased. Dextranase had a significant inhibitory effect on dental plaques.

#### 2.3.6. Porous Starch of Sweet Potato

The samples were observed using SEM, and the starch was shown to be porous gradually by increasing the incubated time (Figure 8). The surface of the starch was smooth and intact at 0 h. After 3 h, the starch surface became depressed, forming mostly uniform pores that penetrated from the surface inwardly. The crater-like structure of the starch surface increased with time. After 15 h of incubation, the pores could be observed in the starch. The pores collapsed and there was a greater degree of granule porous from the surface inward. The results showed that the dextranase was able to prepare porous starch of sweet potato.

Water and oil absorption by sweet potato porous starch increased significantly with the prolongation of enzymatic digestion (Figure 9). Water and oil absorption reached 80% and 78%, respectively, after the starch was digested for 15 h by dextranase. The concaves and pores on the starch surface considerably increased the surface area of the starch particles, which made the entry of water and oil molecules into the interior of the starch easier and improved the absorption [33].

## 3. Discussion

Dextranases play a pivotal role in human health and are used in oral health to effectively treat dental plaques by restraining the biofilm formed by *S. mutans*, but finding the right dextranase to use in daily oral care has, so far, been a challenge [34]. In gut health, the dextranase hydrolysis of dextrose produces highly polymerized iso-oligosaccharides that act as prebiotics to promote the growth and multiplication of intestinal probiotics, thus enhancing intestinal ecology to prevent inflammation [35]. Due to its marine origin, dextranase has better alkali and salt tolerance traits and wider commercial application prospects than terrestrial dextranase [36].

We ultrafiltered and characterized dextranase from the marine-derived *Saccha-romonas* sp. K1. To the best of our knowledge, dextranase-producing marine-originated actinomycetes have not been reported. Actinomycetes strains were more alkali- and salt-tolerant than fungi. The molecular weights of dextranases found so far ranged from 23 kDa to 114 kDa [25,34,35]. By contrast, the molecular weight of dextranase produced by *Saccharomyces* sp. K1 was approximately 9.5 kDa. Furthermore, the obtained dextranase was enzymatically active under both neutral and alkaline conditions (pH 6–11), with an optimum pH of 8.5 and a good stability [13,19]. Additionally, metal ions such as Mg^2+^, Ba^2+^, and Zn^2+^ had only a slight effect on dextranase enzyme activity. Marine dextranase has recently been reported to be effective in removing plaque [34,37], and at least 15 U/mL of *Bacillus aquimaris* PX02 dextranase was required to achieve 90% inhibition of dental plaque biofilm [38]. When the dextranase of *Bacillus hydrophila* S5 was 8 U/mL, the inhibition rate was 80% [34].

The enzymes that can hydrolyze (1–4) and (1–6) glycosidic bonds are available for preparing porous starch. Hui You et al. used a complex amylase to prepare rice porous starch [39]. Mohan Das used pullulanase to prepare banana porous starch [40]. Angela Dura investigated the preparation of porous starch from maize starch by using cyclodextrin glycosyltransferase (EC 2.4.1.19) [41]. Yaiza Benavent-Gil prepared porous starches by treating wheat, rice, potato, and cassava starches with amyloglucosidase, alpha-amylase, and cyclodextrin-glycosyltransferase [42]. Purwadi investigated the effect of the size and solid content of sweet potato starch in the presence of enzymes by using endogenous β-amylase for hydrolyzing the starch [43]. Our experiments provide a new application for dextranase in preparing the porous starch of sweet potato.

Porous starches have received increasing attention because of their functionality and applications, and they can be obtained using physical, chemical, enzymatic, and synergistic methods. Enzyme catalysis is the most effective method [44,45]. Porous starches have been used to encapsulate unstable food ingredients and pharmaceuticals and adsorb heavy metals. The application of dextranase for preparing multi-functional porous starch deserves further study.

## 4. Materials and Methods

### 4.1. Materials

Sea mud samples were collected from Haizhou Bay, Jiangsu, China. Dextran (T20, T40, T70, and T500) and blue dextran 2000 was obtained from GE Healthcare (Uppsala, Sweden). All other reagents of the highest analytical grade were purchased from Sinopharm Chemical Reagent Corp. (Shanghai, China).

### 4.2. Methods

#### 4.2.1. Screening of Dextranase-Producing Strains

Actinomycetes for dextranase production were screened for using blue-dextran medium [46]. The medium contained 1 g of yeast extract, 5 g of peptone fish powder, 2 g of blue dextran 2000, 18 g of agar powder, and 1 L of aged sea water and had an initial pH of 8.0. A suspension was prepared through the gradient dilution of the sample with an appropriate amount of sterile saline. The suspension was spread uniformly on a blue dextran plate. The plates were incubated at 30 °C for 1 week at a constant temperature. The isolate that produced the highest ratio of a transparent zone to colony diameter was selected for further study.

Actinomycete isolation and purification was performed using the modified Koss No. 1 medium [47]. The medium contained 20 g of soluble starch, 0.5 g of NaCl, 0.01 g of FeSO_4_, 1 g of KNO_3_, 0.5 g of K_2_HPO_4_, 0.5 g of MgSO_4_, 1 g of K_2_Cr_2_O_7_, and 1 L of aged sea water, with an initial pH of 8.0.

#### 4.2.2. Identification of Strain K1

The phenotypic characteristics of strain K1 were identified by observing the colony morphology and the cells through light microscopy and scanning electron microscopy (SEM, JFC-1600 type, JSM-6390LA; JEOL, Tokyo, Japan). The slides inserted into the medium were removed after 7 days of incubation at 30 °C. After the samples were dried and sprayed with gold, they were observed through SEM [48,49]. After the strains were incubated at 30 °C for 3 days, DNA from the cells was extracted using a gene extraction kit (TianGen, Beijing, China). The 16S rDNA sequence of the strains was amplified through PCR by using primers 27F (5′-AGAGTTTGATCCTGGCTCAG-3′) and 1492R (5′-GGTTACCTTGTTACGACTT-3′). The PCR reaction system (50 μL), 5× Q5 Reaction Buffer 10 μL, 10 mM dNTPs 1 μL, 10 μM Forward Primer 2.5 μL, 10 μM Reverse Primer 2.5 μL, template DNA 2 μL, Q5 High-Fidelity DNA Polymerase 0.5 μL, dd H_2_O 31.5 μL. The amplification conditions for 16S rDNA were initial denaturation at 94 °C for 3 min, followed by 35 cycles of denaturation at 94 °C for 30 s, annealing at 60 °C for 30 s, and extension at 72 °C for 2 min. The final extension was performed at 72 °C for 5 min. The PCR products were sent to Sangon (Shanghai, China) for sequencing, and the sequences were searched for correlation by using NCBI (https://www.ncbi.nlm.nih.gov/, accessed on 28 April 2022).

#### 4.2.3. Fermentation of Dextranase

The initial fermentation medium of dextranase contained 1 g/L of yeast extract, 5 g/L of fish powder peptone, and 10 g/L of dextran T20 (pH 8.0). The seed medium (pH 8.0) was the same, except it did not contain dextran T20. Seeds were cultured for 24 h, inoculated into the fermentation medium, and incubated at 30 °C and 180 rpm for 48 h. The broth was centrifuged at 8000 rpm for 5 min, and then the supernatant was ultrafiltered using 8 kDa Millipore (Darmstadt, German) and used later. The activity of dextranase was monitored using the 3,5-dinitrosalicylic acid method. Briefly, the dextranase was incubated with an equal amount of 2% dextran T20 dissolved in a 50 mM pH 5.5 acetic acid buffer at 50 °C [49]. One unit of dextranase activity is defined as the amount of enzyme that produces 1 µmol of reducing sugar per minute under the aforementioned conditions.

#### 4.2.4. Conditions for Dextranase Production

Different carbon sources (1 g/L; maltose, D-levulose, soluble starch, yeast extract, glucose, lactose, and dextrin) and nitrogen sources (5 g/L; tryptone, peptone fish powder, ammonium chloride, ammonium sulfate, bean pulp, nitrate of potash, and sodium nitrate) were selected to replace the yeast extract or peptone in the fermentation medium. The effects of these carbon and nitrogen sources on the dextranase activity were compared with the relative enzymatic activity.

To determine the optimal incubation time and inoculum amount of the seed solution, the solution was incubated for 24–96 h. Then, 5–10% of the seed solution was inoculated in the fermentation medium. Samples were collected after 48 h to determine the optimal incubation time and optimal inoculum amount. Then, the fermentation medium was cultured at 30 °C and 180 rpm for 6 days to monitor the dextranase activity. The enzyme activity was measured every 12 h.

The effects of temperature on dextranase production were assessed by culturing strain K1 from 28 to 40 °C for 72 h. To determine the optimal initial pH for dextranase production, different initial pH values (6–11) of the medium were selected and incubated at 40 °C for 72 h. Subsequent experiments were conducted under the optimal temperature and initial pH conditions.

Different volumes of the medium (15–45 mL) were added to 100 mL Erlenmeyer flasks to test the effect of liquid content on the dextranase production. Dextran of different molecular weights (T20, T40, T70, and T500) was selected as an inducer for dextranase production, and then, different dextran concentrations (0–1.6%) were added to detect the effects on dextranase production.

#### 4.2.5. Enzymatic Properties of Dextranase

In order to determine the effect of pH on the activity of dextranase, the following different pH (50 mM) buffers were configured, phosphate buffer (pH 6–7.5), Tris-HCl (pH 7.5–9), and glycine-NaOH (pH 9–11). The dextranase was mixed with the different buffers. The enzymatic activity was measured at 50 °C. Then, the mixture was incubated for 1 h at 25 °C, and the residual enzyme activity was detected to determine the enzyme stability.

To determine the effect of temperature on the enzyme, the dextranase was added to 50 mM buffer (pH 8.5) and the enzyme activity was determined from 25 °C to 60 °C. To determine the thermal stability, dextranase was preheated for 1 to 5 h at different temperatures (45 °C, 50 °C, 55 °C, and 60 °C).

The molecular weight of dextranase was determined using the method below. The dextranase was ultrafiltered using an 8-kDa Minimate^TM^ TFF capsule with an area of 50 cm^2^ and a Pall Minimate^TM^ TFF system (Pall Corporation, Port Washington, NY, USA) under 30 psi pressure. The dextranase was concentrated 20-fold. The concentrated dextranase was combined with bromophenol blue under a boiling water bath. The molecular weight of dextranase was detected by Tricine-SDS-PAGE according to the method of Lai et al. [25]. Dextranase in gels was detected by a 15% Tricine-SDS-PAGE gel containing 0.5% blue dextran. The protein markers were stained with Komas Brilliant Blue after the transparent band appeared under the treatment of Triton-100 50% reversion in a pH 8.5 buffer at 50 °C for 2 h. The molecular weight of the transparent band was observed after decolorization.

The enzyme solution was prepared via centrifugation (8000 rpm, 5 min) and ultrafiltration. The Millipore had a molecular weight cut-off of 8 kDa. The solution over 8 kDa was collected and the dextranase activity was detected using the 3,5-dinitrosalicylic acid (DNS) method.

To determine the effect of metal ions on the dextranase activity, chloride salts (Ca^2+^, Ba^2+^, Mg^2+^, Fe^3+^, Co^2+^, Sr^2+^, and K^+^) and sulfate (Zn^2+^) were used. The dextranase was added to a buffer containing different concentrations of metal ions (1, 5, and 10 mM).

The substrate specificity of dextranase was investigated. The dextranase was mixed with substrates with different glucosidic linkages at 50 °C for 15 min to measure the enzyme activity.

#### 4.2.6. Analysis of Hydrolysates

The hydrolysates of dextranase were analyzed using a high-performance liquid chromatography (HPLC) system (Waters 600, Milford, MA, USA). Then, 500 μL of dextranase was mixed with 1.5 mL of 3% dextran T20 at 40 °C for 1, 2, 6, 12, and 24 h. The solution was then boiled for 10 min to terminate the reaction, and the supernatant was centrifuged at 12,000 rpm for 5 min and filtered through a 0.45 μm filter. The products were analyzed using the column of Waters Sugar-Pak1 (6.5 × 300 mm; Waters, Milford, MA, USA). The mobile phase was pure water at a flow rate of 0.4 mL/min. The column temperature was 75 °C and the injection volume of the sample was 20 μL [27]. Information was handled using Engage GPC programming (Waters, Milford, MA, USA). Quantification of the oligosaccharide composition was performed using the peak area method.

#### 4.2.7. Biofilm Removal by Dextranase

Biofilms were detected according to the Tian Deng’s protocol [23]. The minimum biofilm inhibition concentration (MBIC) was detected. *Streptococcus pyogenes* was first activated in sucrose-free brain heart infusion (BHI) medium at 37 °C for 18 h. Activated *S. pyogenes* ATCC 25175 was incubated anaerobically for 24 h at 37 °C in BHI (1% sucrose) with different concentrations of dextranase (0, 2, 4, 6, 8, and 10 U) in 96-well plates. Then, the medium was removed and the cells were rinsed with sterile water three times. The plates were dried at room temperature for 1 h. The cells were stained with 1% crystal violet solution for 5 min. The staining solution was removed and 200 µL of 95% ethanol was added. The optical density was measured at 595 nm by using a plate spectrophotometer (Bio-Rad, Hercules, CA, USA). Then, the biofilm inhibition rate was calculated. Inhibition rate (%) = (1 − experimental group/control group) × 100.

The minimum biofilm reduction concentration (MBRC) of *S. pyogenes* ATCC 25175 was assessed in the same manner as the MBIC. The MBRC was different, because dextranase was added after *S. pyogenes* ATCC 25175 formed the biofilm (24 h later).

To removal dental plaque biofilm, a sterile coverslip was placed in a 24-well plate, and *S. mutans* was added with BHI medium at 1 × 10^8^ cfu/mL and incubated anaerobically at 37 °C for 24 h. Then, the bacterial liquid was aspirated and washed with 400 μL of sterile water. BHI medium containing dextranase (0, 2, 4, 6, 8, and 10 U) was added and incubated anaerobically at 37 °C for 6 h. The medium was discarded. The 2.5% glutaraldehyde was added to each well and the plates were placed in a refrigerator for 4 h. Afterwards, the biofilm was dehydrated for 15 min using 40%, 60%, 80%, and 100% alcohol sequentially for each concentration. Gold was sprayed before the biofilm was observed through SEM [38].

#### 4.2.8. Preparation of Porous Starch and Determination of Water and Oil Absorption

The sweet potato starch (Rong Fang, Beijing, China) with a high content of (1–6) glycosidic bonds was selected. Three grams of starch were weighed and mixed with 15 mL (pH 8.5) of Tris-HCl buffer, then transferred to a shaker and preheated for 30 min at 50 °C. After preheating, the appropriate amount of enzyme solution was added. The solution was incubated at 50 °C for different times (0–15 h). The reaction was terminated by adding ethanol, and the precipitate was removed through centrifugation. The samples were dried and ground. The samples were observed under a scanning electron microscope.

Liu’s method [50] was used to determine the water and oil absorption of porous starch. In total, 1 g of enzyme-treated sweet potato starch and natural sweet potato starch was weighed in a small beaker, 2 mL of peanut oil/water was added and stirred for 30 min, poured into a centrifuge tube of a known mass, and centrifuged at 4000 r/min for 10 min. The centrifuged supernatant layer was poured out and then inverted until the oil and water no longer oozed out. The oil absorption rate was calculated according to the formula: Oil/water absorption rate (%) = (mass of starch after centrifugation and water absorption by centrifuge tubes − dry mass of porous starch − mass of centrifuge tubes)/ dry mass of porous starch × 100%.

#### 4.2.9. Data Analysis

All experiments included three parallel samples, and the data were analyzed using SPSS v19 software. The graphs were drawn using Origin software (2019, OriginLab, Northampton, MA, USA).

## 5. Conclusions

In this study, dextranase-producing strain K1 was screened from sea mud. The microbe was identified as a marine actinomycete *Saccharomonospora* sp. based on the 16S rDNA gene sequence and its morphology. The enzyme production conditions were optimized, and the optimum carbon and nitrogen sources were soluble starch and fish powder, respectively. The optimal fermentation time, temperature, initial pH, NaCl concentration, and inducer concentration were 72 h, 37 °C, 8.5, 6 g/L, and 10 g/L, respectively. The enzyme activity could be maintained at 80% by holding at 50 °C for 5 h. The enzyme stability was better at pH 7.5–9.5. The molecular weight of dextranase was approximately 9.5 kDa. The main components of the hydrolysate were the highly polymerized isomaltopentaose and isomaltoheptose. Moreover, dextranase exhibited significant efficacy in the removal of dental plaque. Meanwhile, dextranase can be used for preparing sweet potato porous starch, which can be employed as a drug carrier. 

## Figures and Tables

**Figure 1 marinedrugs-22-00069-f001:**
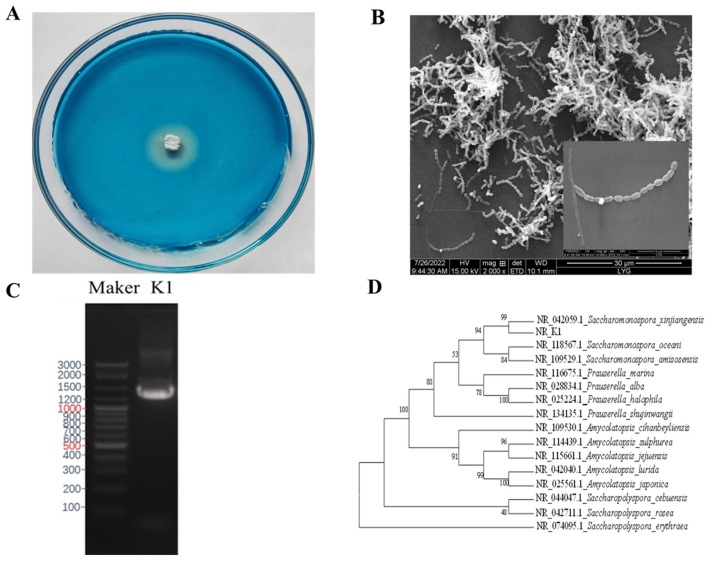
Clear zone formed by strain K1 on the blue dextrose plate (**A**) and scanning electron micrograph (**B**). Agarose electrophoresis profiles of 16S rDNA PCR products M: DNA marker (Sangon, Shanghai, China); strain K1 rDNA PCR amplification product (**C**). Phylogenetic tree based on 16S rDNA gene sequences (**D**).

**Figure 2 marinedrugs-22-00069-f002:**
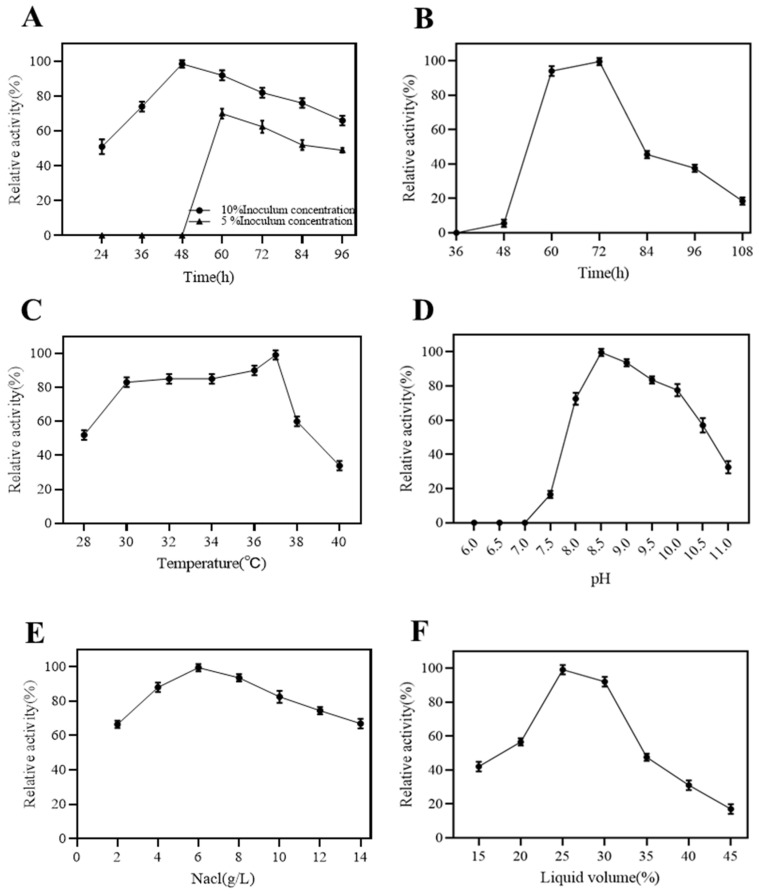
The dextranase produced by strain K1 was affected by inoculum and culture time (5% and 10%, 24–96 h) (**A**), fermentation time (36–108 h) (**B**), culture temperature (28–40 °C) (**C**), pH (6–11) (**D**), NaCl concentration (2–14 g/L) (**E**), and liquid volume in Erlenmeyer flask (15–45%) (**F**).

**Figure 3 marinedrugs-22-00069-f003:**
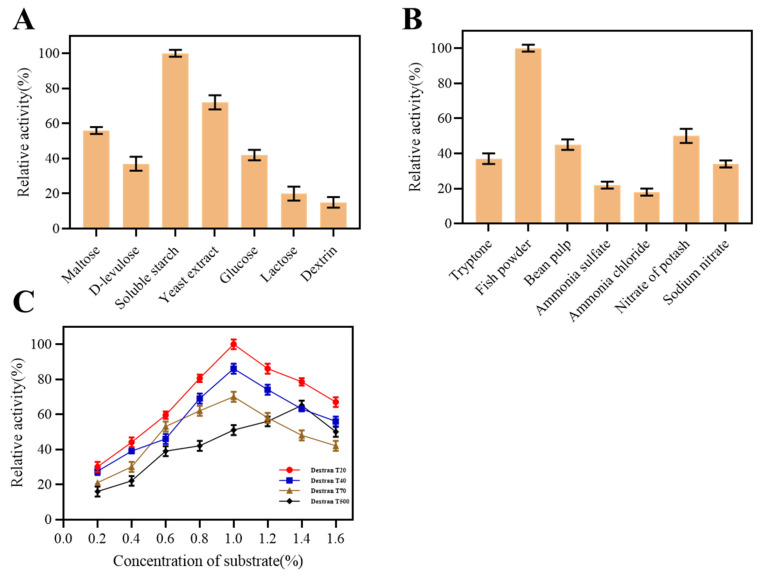
The dextranase activity produced by strain K1 was affected by carbon source (maltose, D-levulose, soluble starch, yeast extract, glucose lactose, and dextrin) (**A**), nitrogen source (tryptone, fish powder, bean pulp, ammonia sulfate, ammonia chloride, nitrate of potash, and sodium nitrate) (**B**), and inducers (dextran T20, T40, T70, and T500) and concentrations of inducers (0.2–1.6%) (**C**).

**Figure 4 marinedrugs-22-00069-f004:**
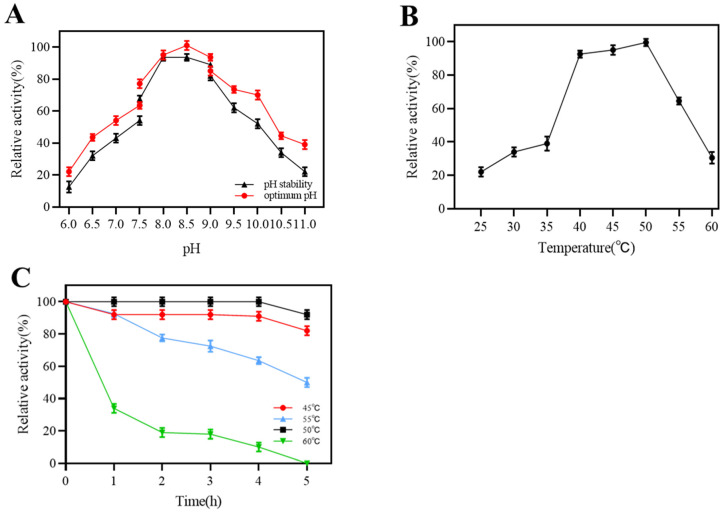
Effects of pH (6–11) (**A**), temperature (25–60 °C) (**B**), and incubation at 45–60 °C (**C**) on dextranase activity and stability. The highest activity was set as 100% to calculate the relative activity.

**Figure 5 marinedrugs-22-00069-f005:**
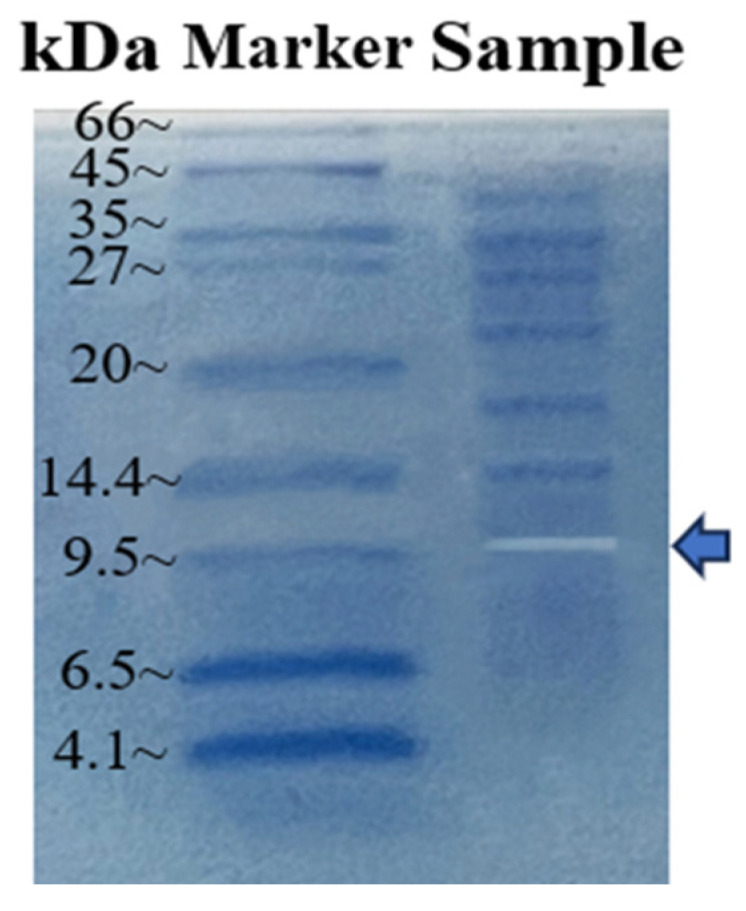
M Molecular weight standard (BBI^®^, C600201) A transparent bar showing dextranase activity on a 15% native Tricine-SDS-PAGE gel containing 0.5% (*w*/*v*) blue dextran, and the number on the left is the size of the molecular weight of the labeled protein.

**Figure 6 marinedrugs-22-00069-f006:**
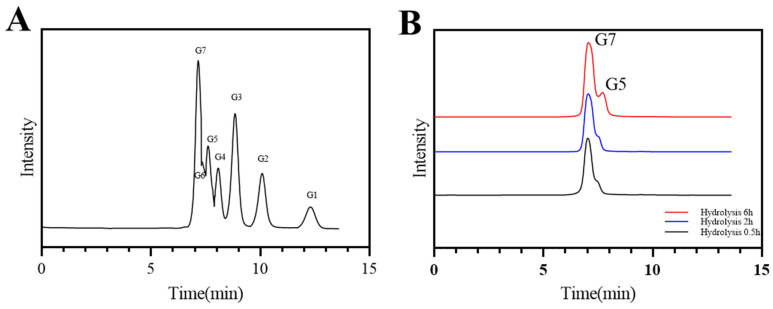
Products of the reaction between dextranase and 3% dextran T20 at 50 °C for different times determined through high-performance liquid chromatography: (**A**) standard results (G1 to G7) for glucose, maltose, maltotriose, maltotetraose, maltopentaose, maltohexaose, and maltoheptaose standard sugars, and (**B**) results of dextranase hydrolysis to 3% dextran T20 for 0.5 to 6 h.

**Figure 7 marinedrugs-22-00069-f007:**
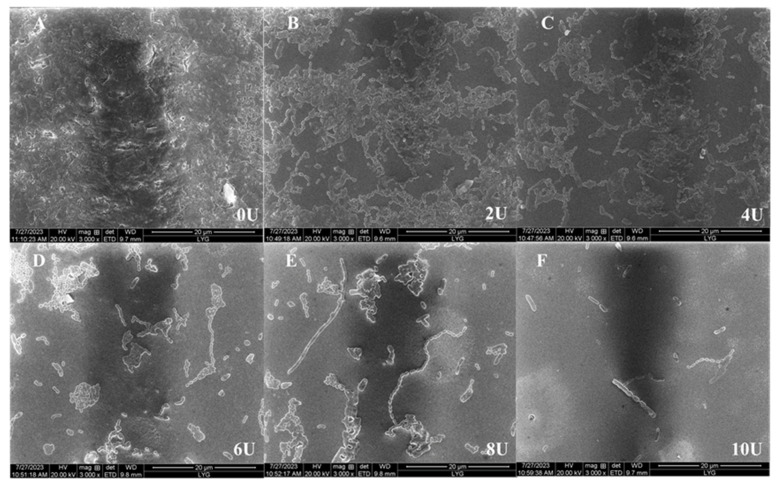
Effect of the dextranase on the dental plaque biofilm formed by *S. mutans* (3000×); The control group was 0 U/mL, and deionized water with the same volume of sterile water was used instead of dextranase. The presence of dextranase at the concentration of (**A**) 0, (**B**) 2, (**C**) 4, (**D**) 6, (**E**) 8, and (**F**) 10 U/mL.

**Figure 8 marinedrugs-22-00069-f008:**
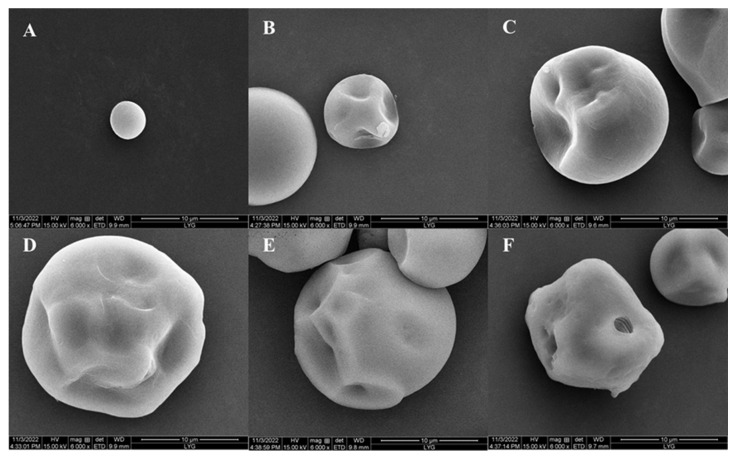
Scanning electron microscopy results of dextranase preparation of porous starch of sweet potato: (**A**) raw starch, (**B**) 3 h, (**C**) 6 h, (**D**) 9 h, (**E**) 12 h, and (**F**)15 h.

**Figure 9 marinedrugs-22-00069-f009:**
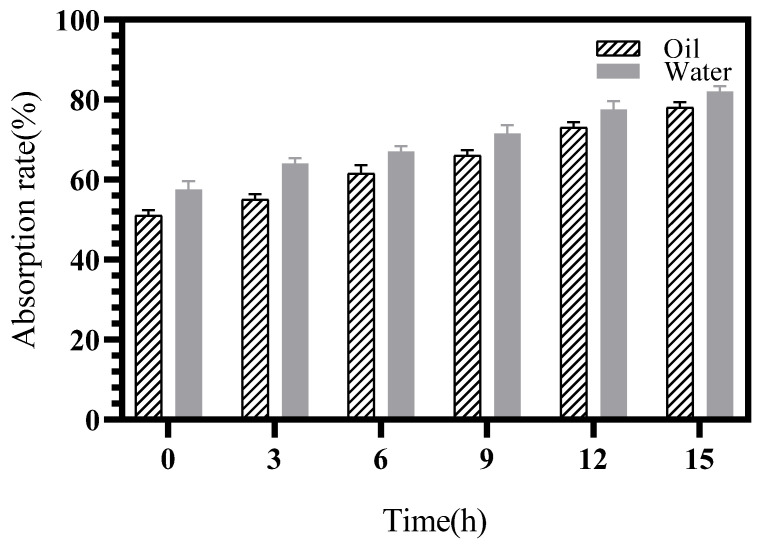
Water and oil absorption rate of sweet potato porous starch.

**Table 1 marinedrugs-22-00069-t001:** Effect of metal ions on dextranase.

Reagents	Relative Activity (%)		
	1 mM	5 mM	10 mM
Control	100.00 ± 0.08	100.00 ± 0.08	100.00 ± 0.08
Ca^2+^	92.40 ± 0.56	102.42 ± 0.24	108.00 ± 0.42
Ba^2+^	90.21 ± 0.08	85.42 ± 0.21	81.52 ± 0.06
Mg^2+^	90.00 ± 0.12	86.21 ± 0.54	80.24 ± 0.24
Co^2+^	75.42 ± 0.48	50.21 ± 0.06	20.84 ± 0.56
Sr^2+^	95.46 ± 1.02	105.26 ± 0.82	124.25 ± 0.54
Zn^2+^	85.24 ± 0.82	72.21 ± 0.24	50.56 ± 0.42
K^+^	98.40 ± 2.20	104.10 ± 1.52	112.62 ± 0.20
Fe^3+^	72.32 ± 0.47	0.00	0.00

**Table 2 marinedrugs-22-00069-t002:** Action of dextranase on diverse carbohydrates.

Substrate	Main Linkages	Relative Activity (%)
Dextran T20	α-1,6	100.00 ± 0.25
Dextran T40	α-1,6	95.00 ± 0.39
Dextran T70	α-1,6	90.00 ± 0.72
Dextran T500	α-1,6	85.00 ± 0.46
Dextran T2000	α-1,6	80.00 ± 0.52
Soluble starch	α-1,4; α-1,6	32.00 ± 0.45
Chitin	β-1,4	0

**Table 3 marinedrugs-22-00069-t003:** Proportion of products of hydrolyzed dextran.

Time of Hydrolysis (h)	Hydrolysates (%)			
	Isomaltoheptaose	Isomaltopentaose	Maltose	Glucose
0.5	98.02	1.81	0.06	0.10
2	97.18	2.18	0.52	0.12
6	72.71	22.43	0.53	4.33

**Table 4 marinedrugs-22-00069-t004:** Biofilm inhibition rates for different dextranase.

Concentration of Dextranase (U/mL)	Biofilm Formation Inhibition Rate (%)	Formed Biofilm Reduction Rate (%)
0	0.00	0.00
2	27.32 ± 0.56	50.12 ± 2.34
4	52.15 ± 2.24	65.14 ± 1.21
6	75.82 ± 1.25	73.28 ± 0.35
8	83.26 ± 2.12	79.24 ± 2.41
10	94.23 ± 0.22	92.54 ± 1.42

## Data Availability

The interaction data used to support the study findings are included within the article. Moreover, all the data used to support the study findings are available from the corresponding author upon request.

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
