# Peer review of "The Screening and Identification of a Dextranase-Secreting Marine Actinmycete Saccharomonospora sp. K1 and Study of Its Enzymatic Characteristics"

_marinedrugs, 2024, doi:10.3390/md22020069_

Round 1
Reviewer 1 Report (Previous Reviewer 1)
Comments and Suggestions for Authors
After reviewing the revisions in the article, the response to recommendation 1 is acceptable.
For recommendation #2, the authors have added more detail to the section 4.2.5. However, ultrafiltration is not an acceptable purification method of the dextranase because the SDS page gel in Figure 5 still shows many bands between 9.5 and 66kDa. The recommendation was to use a chromatography method such as gel filtration to purify the dextranase before MW determination.
For recommendation #3, Figure 6B still shows the minor peak labelled as G6. Since the retention times of G7, G6 and G5 are relatively close, one could try to improve the separation method to resolve the peaks and/or try a co-injection of the pure standards with the hydrolysis product to determine which products co-elute.
Author Response
Dear Editor and Reviewers,
We are really appreciated for your kindly consideration to give us an opportunity to revise our manuscript entitled “Screening and identification of a dextranase-secreting marine actinomycete Saccharomonospora sp. K1 and study of its enzymatic characteristics” (Marine drugs-2748271). We appreciate reviewers for the comments, and we have amended our manuscript according to the comments carefully. We have tried our best to revise our manuscript. Revised portions have been marked in yellow background in the manuscript. The main corrections in the paper and the responds to the reviewer’s comments are as following.
Responds to the reviewer 1’s comments:
- For recommendation #2, the authors have added more detail to the section 4.2.5. However, ultrafiltration is not an acceptable purification method of the dextranase because the SDS page gel in Figure 5 still shows many bands between 9.5 and 66kDa. The recommendation was to use a chromatography method such as gel filtration to purify the dextranase before MW determination.
Response: Thank you very much for your comments. Yes, we are agreed with you about the purification. However, we are focused on the application of dextranase, and the methods was used in the dextranase purification is referred on methods of company. In figure 5, the enzyme solution was purified by centrifuge (8000 rpm, 5 min) and ultrafiltration (8 kDa). So, the bands will be a few between 9.5 and 66 kDa. Thanks.
- For recommendation #3, Figure 6B still shows the minor peak labelled as G6. Since the retention times of G7, G6 and G5 are relatively close, one could try to improve the separation method to resolve the peaks and/or try a co-injection of the pure standards with the hydrolysis product to determine which products co-elute
Response: Thanks for your comment. We have revised in the manuscript. Thanks.
We would like to express our great appreciation to you for comments that make our manuscript improved.
Yours sincerely,

Reviewer 2 Report (Previous Reviewer 2)
Comments and Suggestions for Authors
This paper describes the screening of dextranase producing marine bacterial strain, characterization of the enzyme, and application of the enzyme for biofilm removal and porous starch production. There are some parts where the previous comments were not properly addressed, and there are also some careless mistakes, so I hope that the authors will pay close attention to the revised manuscript.
As I wrote in the previous comment, it is common knowledge to use the purified enzyme when investigating the enzymatic properties. However, in this paper, the authors are examining various properties of enzymes using that have only been ultrafiltrated, so it is necessary to write an explanation between sections 2.3 and 2.3.1
I pointed out “in figures showing the results of studies on culture conditions (Fig. 2 and 3), conditions other than the parameters examined should be attached to each legend” in my previous comment, but the current legends don’t mention them, so the author should revise them. And, the text in the explanation in Fig. 2A is too small to read. Please make it bigger or explain the symbols in the legend.
Furthermore, I also pointed out “In 4.2.5 and 4.2.7, colons can be seen in the sentences, but these are not considered appropriate for academic papers” in my previous comment. In this revised manuscript, the colons were simply deleted; that parts are not part of the sentences. I don’t think this is appropriate for an academic paper.
Fig. 4D uses the notation “mg” to indicate the kinetics, but since this preparation contains a large amount of protein other than the target enzyme, this data is meaningless and should be deleted.
Minor points
Line 116: Figure 4D → Figure 5
Lines 271 and 282: The name of titles are the same.
There is a high possibility that there may have been other careless mistakes, so please review carefully before sending the new revised version.
Comments on the Quality of English Language
As I wrote I the comments, there are some parts that violate the format of an academic paper, so I think it needs to be revised.
Author Response
Dear Editor and Reviewers,
We are really appreciated for your kindly consideration to give us an opportunity to revise our manuscript entitled “Screening and identification of a dextranase-secreting marine actinomycete Saccharomonospora sp. K1 and study of its enzymatic characteristics” (Marine drugs-2748271). We appreciate reviewers for the comments, and we have amended our manuscript according to the comments carefully. We have tried our best to revise our manuscript. Revised portions have been marked in yellow background in the manuscript. The main corrections in the paper and the responds to the reviewer’s comments are as following.
Responds to the reviewer 2’s comments:
- As I wrote in the previous comment, it is common knowledge to use the purified enzyme when investigating the enzymatic properties. However, in this paper, the authors are examining various properties of enzymes using that have only been ultrafiltrated, so it is necessary to write an explanation between sections 2.3 and 2.3.1
Response: Thanks for your comment. we have revised in the manuscript. Thanks.
- I pointed out “in figures showing the results of studies on culture conditions (Fig. 2 and 3), conditions other than the parameters examined should be attached to each legend” in my previous comment, but the current legends don’t mention them, so the author should revise them. And, the text in the explanation in Fig. 2A is too small to read. Please make it bigger or explain the symbols in the legend.
Response: Thank you very much for your comments. We have revised in the manuscript. Thanks.
- Furthermore, I also pointed out “In 4.2.5 and 4.2.7, colons can be seen in the sentences, but these are not considered appropriate for academic papers” in my previous comment. In this revised manuscript, the colons were simply deleted; that parts are not part of the sentences. I don’t think this is appropriate for an academic paper.
Response: Thanks for your comment. We have revised in the manuscript. Thanks.
- 4D uses the notation “mg” to indicate the kinetics, but since this preparation contains a large amount of protein other than the target enzyme, this data is meaningless and should be deleted.
Response: Thanks for your comment. We have revised in the manuscript. Thanks.
- Minor points,Line 116: Figure 4D → Figure 5,Lines 271 and 282: The name of titles are the same.
Response: Thanks for your comment. relevant content has been changes made to the article We have revised in the manuscript. Thanks.
We would like to express our great appreciation to you for comments that make our manuscript improved.
Yours sincerely,

Round 2
Reviewer 1 Report (Previous Reviewer 1)
Comments and Suggestions for Authors
The revised manuscript is suitable for publication after some minor revisions:
1. lines 208-209: please review references 27, 41 and 42. There is no data for the molecular weights of dextranase between MW 23kDa-114kDa. Ref 41 70kDa and Ref 42: 58KDa. Reference 27 has no MW data.
2. Please consider changing the word "purified" on line 325. The protein was concentrated by ultrafiltration, not purified and it afforded a crude preparation based on SDS-page analysis. For examples of protein purification methods please consult references 41 and 42.
Author Response
Dear Editor and Reviewers,
We appreciate reviewers for the comments, and we have amended our manuscript according to the comments carefully. Revised portions have been marked in yellow background in the manuscript. The main corrections in the paper and the responds to the reviewer’s comments are as following.
Responds to the reviewer 2’s comments:
- lines 208-209: please review references 27, 41 and 42. There is no data for the molecular weights of dextranase between MW 23kDa-114kDa. Ref 41 70kDa and Ref 42: 58KDa. Reference 27 has no MW data.
Response: Thank you very much for your comment. We have change reference 27 to 33. Ref 33: 23kDa -114 kDa. We have revised in the manuscript. Thanks.
- Please consider changing the word "purified" on line 325. The protein was concentrated by ultrafiltration, not purified and it afforded a crude preparation based on SDS-page analysis. For examples of protein purification methods please consult references 41 and 42.
Response: Thank you very much for your comments. We have revised as follow in the manuscript. Thanks.
“The enzyme solution was prepared by centrifuge (8000 rpm, 5 min) and ultrafiltration.”
We would like to express our great appreciation to you for comments that make our manuscript improved.
Yours sincerely,

Reviewer 2 Report (Previous Reviewer 2)
Comments and Suggestions for Authors
Most of the points we pointed out have been revised. However, the legends in Fig. 2 and 3 have not been improved. IN Fig. 2 and 3, the authors investigated the culture conditions, so they should be the same except for the parameters that were changed. Please state the conditions (basic conditions other than the parameters to be considered).
Comments on the Quality of English Language
There are some parts that violate the format of an academic paper, so I think it needs to be revised.
Author Response
Dear Editor and Reviewers,
We appreciate reviewers for the comments, and we have amended our manuscript according to the comments carefully. Revised portions have been marked in yellow background in the manuscript. The main corrections in the paper and the responds to the reviewer’s comments are as following.
Responds to the reviewer 2’s comments:
- Most of the points we pointed out have been revised. However, the legends in Fig. 2 and 3 have not been improved. IN Fig. 2 and 3, the authors investigated the culture conditions, so they should be the same except for the parameters that were changed. Please state the conditions (basic conditions other than the parameters to be considered).
Response: Thank you very much for your comment. We have revised in the manuscript.
- There are some parts that violate the format of an academic paper, so I think it needs to be revised.
Response: Thank you very much for your comments. The manuscript has been edited by LEXIS Academic Editing Service to improve English. Thanks.
We would like to express our great appreciation to you for comments that make our manuscript improved.
Yours sincerely,

This manuscript is a resubmission of an earlier submission. The following is a list of the peer review reports and author responses from that submission.
Round 1
Reviewer 1 Report
Comments and Suggestions for Authors
The authors have described the identification and characterization of a dextranase enzyme from a marine microorganism. The manuscript is generally well written but some clarification is required on a few items:
1. line 100. "Most dextranases from the sea have an optimum pH of 8.0, with most of them being acidic....." This is contradictory information. Please clarify if you are referring to terrestrial dextranases having a low pH range. Also provide a reference that supports the statement that most sea dextranases have a basic pH optimum.
2. line 116. The molecular weight is estimated from a denaturing SDS page gel and is significantly lower than the MW of dextranases reported in the literature. This could be the MW of the monomeric unit, if the protein is a dimer. The MW needs to be confirmed by native gel conditions and/or gel filtration chromatography to rule out degradation of the protein under denaturation conditions.
line 156 Figure 6. (B) How was it determined that G6 was the minor product and not G5? Based on retention times of the stds it looks closer to G5.
Comments on the Quality of English Language
Acceptable
Reviewer 2 Report
Comments and Suggestions for Authors
This paper describes the screening of dextranase producing marine bacterial strain, characterization of the enzyme, and application of the enzyme for biofilm removal and porous starch production. But my honest impression is that there are some deficiencies that make it unsuitable for publication.
First, although it is common knowledge to use the purified enzyme when investigating the enzymatic properties, there is no description of the process for purifying the enzyme. Furthermore, the method for preparing the enzyme used in 4.2.5 is not described. The purity of the enzyme is shown in Fig. 5, but there are many impurities, Therefore, it cannot be denied that the results described in 2.3.2 and 2.3.3 may have been obtained by multiple enzymes contained in this sample.
In addition, in figures showing the results of studies on culture conditions (Fig. 2 and 3), conditions other than the parameters examined should be attached to each legend. In 4.2.5 and 4.2.7, colons can be seen in the sentences, but these are not considered appropriate for academic papers.
Comments on the Quality of English Language
As written in the comments for author, colons can be seen in the sentences, but these are not considered appropriate for academic papers.
Reviewer 3 Report
Comments and Suggestions for Authors
The authors isolate a strain of Saccharomonospora sp. and try to optimize the production of a dextranase-secreting enzyme from this strain.
The main problem identified in this study is the experiment on SDS-PAGE gel that shows a band which is decolorized around 9.5 kDa. I don’t understand why the dextranase stayed active under denaturated conditions in presence of SDS and I am not convinced by the fact that this band at 9.5 kDa could be the dextranase of interest. In the literature, the dextranase have a greater molecular weight.
Moreover, in the materials and methods section, the authors describe the study of Vmax and KM and test a lot of physical and chemical parameters on the activity of their extract (temperature, pH, stability for example) and also characterized the microbicidal effect on their mixture.
The main problem is that the authors work with a mixture and not on a purified enzyme. It is not demonstrated that it is the action of a dextranase on biofilm for example and not another molecules in the mixture.
Moreover, some Saccahromonospora strain have been completely sequenced (2009 Pati et al. Genomic Science). It could be great if the authors search in database and from bioinformatics studies the open reading frame of putative dextranase and check the molecular weight for example. Or, if the authors will purify the enzyme or cut on gel the band(s) corresponding to the enzyme and analyze by mass spectrometry, they could have an idea of the amino acid sequence and so the molecular weight.
The study needs better proof concerning the dextranase presence in the media and the authors need to improve the characterization of the enzyme from sequence, mass spectrometry and purification point of views.